# Irminger Sea deep convection injects oxygen and anthropogenic carbon to the ocean interior

F. Fröb[1,2], A. Olsen[1,2], K. Våge[1,2], G.W.K. Moore[3], I. Yashayaev[4], E. Jeansson[2,5] & B. Rajasakaren[2,5]

Deep convection in the subpolar North Atlantic ventilates the ocean for atmospheric gases through the formation of deep water masses. Variability in the intensity of deep convection is believed to have caused large variations in North Atlantic anthropogenic carbon storage over the past decades, but observations of the properties during active convection are missing. Here we document the origin, extent and chemical properties of the deepest winter mixed layers directly observed in the Irminger Sea. As a result of the deep convection in winter 2014–2015, driven by large oceanic heat loss, mid-depth oxygen concentrations were replenished and anthropogenic carbon storage rates almost tripled compared with Irminger Sea hydrographic section data in 1997 and 2003. Our observations provide unequivocal evidence that ocean ventilation and anthropogenic carbon uptake take place in the Irminger Sea and that their efficiency can be directly linked to atmospheric forcing.

[1] Geophysical Institute, University of Bergen, Bergen 5007, Norway. [2] Bjerknes Centre for Climate Research, Bergen 5007, Norway. [3] Department of Physics, University of Toronto, Toronto, Ontario, Canada M5S A17. [4] Bedford Institute of Oceanography, Fisheries and Oceans Canada, Dartmouth, Nova Scotia, Canada B2Y 4A2. [5] Uni Climate, Uni Research, Bergen 5007, Norway. Correspondence and requests for materials should be addressed to F.F. (email: friederike.frob@uib.no).

Deep convection is a key process for maintaining the oceanic sink for anthropogenic carbon and ventilating the ocean for atmospheric gases such as $CO_2$ and oxygen[1]. The renewal of oxygen at depth supports aerobic remineralization of organic matter, and biogeochemical processes respond sensitively to changes in the supply rate from the surface ocean[2]. Regions where ocean ventilation and anthropogenic carbon sequestration occur are limited, for example, 85% of the total volume of the deep ocean is ventilated from only 15% of the surface ocean[3]. The subpolar North Atlantic is such a location. However, the processes that govern deep convection there are highly variable and sensitive to changes in large-scale ocean–atmosphere interactions.

In the northern mid-latitudes, strong declines in oxygen between the 1970s and the 1990s were observed in the upper ocean (100–1,000 m). This decline in oxygen likely resulted from reduced exchange between the surface mixed layer and interior ocean, associated with warming and freshening in the upper ocean[4,5]. Furthermore, a series of papers have shown how reduced deep water formation in the subpolar North Atlantic from the mid-1990s to the mid-2000s affected ocean anthropogenic carbon storage rates[6–8]. A recent acceleration in anthropogenic carbon storage rates has been observed in the North Atlantic from 2003 to 2014 compared with the 1989–2003 period, attributed to changes in water mass ventilation as well as increasing atmospheric $CO_2$ concentration[9]. The frequency, duration and intensity of deep water formation in the North Atlantic subpolar gyre region is strongly related to atmospheric variability. In particular, variability associated with the North Atlantic oscillation (NAO) is one of the main drivers for hydrographic property changes in the subpolar North Atlantic on interannual to decadal timescales[10,11].

Despite efforts to elucidate the relationship between atmospheric forcing and ventilation processes, a complete mechanistic understanding is still missing. There are insufficient observational data to determine the impact of atmospheric forcing on mixed layer depth and properties in the subpolar gyre, in particular with regard to ocean ventilation and carbon sequestration. Data collected during summer cruises only provide indirect evidence, and signals that are advected from outside the subpolar gyre are challenging to distinguish from signals that reflect local mechanisms of deep water formation. With the installation of mooring stations and the advent of autonomous sampling systems such as Argo the problem of seasonal biases in sampling is now being alleviated[12,13]. However, the type of sensors that are carried are limited and for the collection of high-quality carbon and tracer data, ship-board measurements are required. Since wintertime conditions are extremely harsh in the subpolar gyre, ship data from the convective season are rare. For example, between 1990 and 2014 the Irminger Sea was occupied by more than 30 research cruises, but only once during the potentially convective season[14].

In the Irminger Sea, deep convection takes place under favourable oceanic and atmospheric conditions[15–21]. Here we present new observational data from a unique cruise in winter 2015 that captured such a deep convective event. These observations document the properties of the deepest mixed layers directly recorded in this region and their impact on oxygen and anthropogenic carbon, providing a link between atmospheric forcing and anthropogenic carbon storage and oxygen re-ventilation in the subpolar gyre.

## Results

### Oxygen and anthropogenic carbon in the water column. Sulfur hexafluoride ($SF_6$) saturation values, here shown in a potential temperature–salinity diagram based on the 2015 Irminger Sea cruise data (Fig. 1a), indicate that strong local convection took place in the Irminger Sea in winter 2014–2015. The depth of the mixed layer is derived from hydrographic profiles using a threshold criterion[22] (see Methods section). The mean density in the winter mixed layer corresponds to the density of the lighter Labrador Sea Water (LSW) class formed in the subpolar gyre after 2000 (ref. 23). $SF_6$ is typically undersaturated in the surface ocean during active water mass formation with saturation values of 86% in the North Atlantic[24]. Here, within the winter mixed layer the $SF_6$ saturation varies strongly between undersaturation and supersaturation, with supersaturated values testifying to the recent formation through convective processes in winter 2014–2015. The highest supersaturated values of $SF_6$ were observed at the base of the mixed layer with a mean saturation of $116.2 \pm 8.2\%$, possibly driven by bubble formation at the surface induced by high wind speeds[25].

Oxygen and anthropogenic carbon concentrations are presented along the 2015 cruise transect across the Irminger Sea with the coast of Greenland close to Cape Farewell to the west and the Reykjanes Ridge to the east (Fig. 1b,c). Anthropogenic carbon concentrations are computed using the transient time distribution (TTD) method[26]. Within the winter mixed layer, they are based on $SF_6$ data. In the less recently ventilated water masses below the mixed layer, dichlorodifluoromethane (CFC-12) measurements provide more reliable estimates[24]. Concentrations of oxygen and anthropogenic carbon are high in the entire winter mixed layer, indicative of its recent ventilation. Below lies an older water mass layer with reduced levels of oxygen and anthropogenic carbon. Towards the bottom of the western Irminger Basin, elevated oxygen concentrations are found, a characteristic feature of Denmark Strait Overflow Water. In combination with Iceland Scotland Overflow Water and LSW from the Labrador and Irminger Seas, this water mass forms North Atlantic Deep Water[27], the key component of the lower limb of the Atlantic Meridional Overturning Circulation.

With respect to atmospheric values in 2015, the surface waters are saturated by 94–98% in oxygen and 90–100% in anthropogenic carbon, respectively. Oxygen undersaturation in the northern high latitudes during the convective season has been described as a consequence of gas exchange lagging behind strong heat loss[28,29].

### Mixed layer depths in the North Atlantic subpolar gyre. Deep convection in winter 2014–2015 was not confined to the Irminger Sea, but occurred across the entire subpolar gyre. The mixed layer depth estimates we present are based on individual Argo profiles from December 2001 to May 2015, and hydrographic station data from April and May cruises to the Irminger and Labrador Seas in 2015. For consistency with earlier studies[20], we use the 2000–2007 base period of February–April Argo data to contextualize the 2015 data. Compared with climatological conditions, wintertime mixed layer depths were exceptionally deep in the entire subpolar gyre in 2015 (Fig. 2a) and the largest deviation from the long-term mean occurred in the Irminger Sea (Supplementary Fig. 1). Here, mixed layers were deeper than 1,400 m in early 2015 (Fig. 2a). While these are the deepest mixed layers directly observed in the Irminger Sea, exceeding the two previous winters by at least 800 m, there is indirect evidence that even deeper convection took place in the Irminger Sea in the early 1990s (ref. 15).

Convectively formed water masses are partly advected from their main formation region in the Labrador Sea to the Irminger Sea with a travel time of about 2 years, where these weakly stratified water masses can precondition the water column for convection[5,15,17,23]. Alternatively, waters originating from the

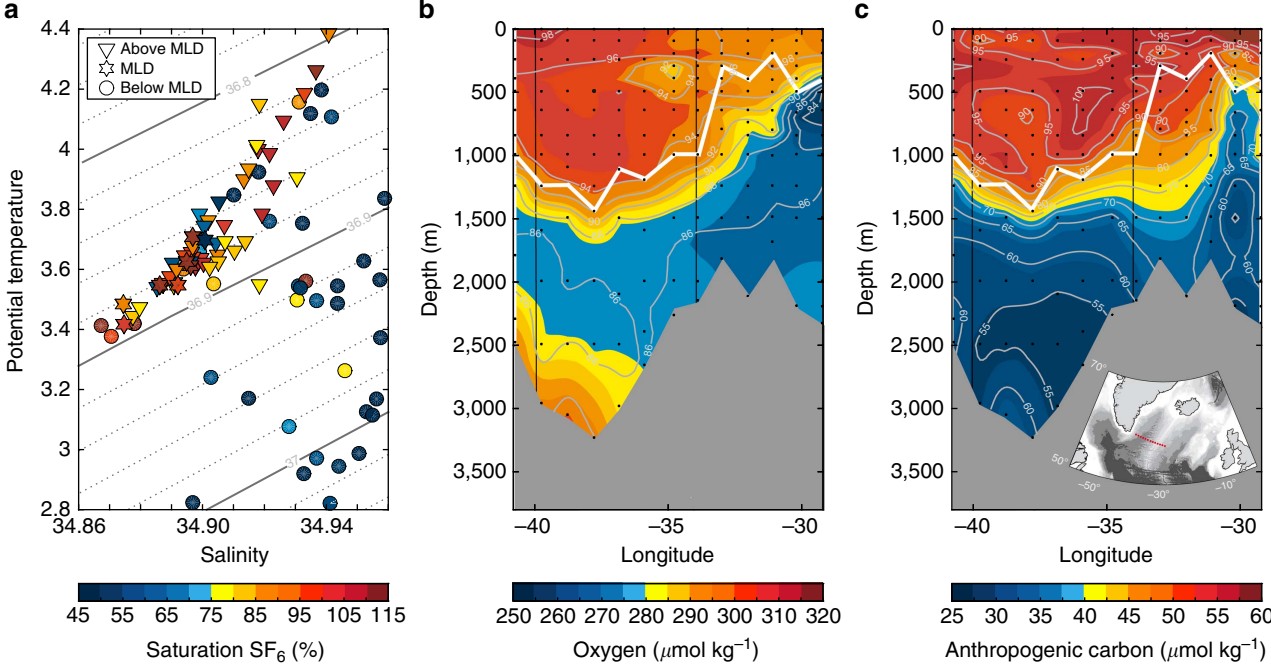

**Figure 1 | Sulfur hexafluoride, oxygen and anthropogenic carbon in the water column in winter 2014–2015.** (**a**) Irminger Sea potential temperature, salinity and sulfur hexafluoride (SF$_6$) saturation as observed during the April Irminger Sea cruise (58GS20150410). The contour lines show the potential density anomaly ($\sigma_2$ isopycnals). The filled circles show data below the depth of the mixed layer (MLD), the filled triangles show data within the winter mixed layer and the stars highlight properties at the base of the winter mixed layer. (**b**) Interpolated oxygen and (**c**) anthropogenic carbon concentrations in μmol kg$^{-1}$ for a transect across the Irminger Sea based on April Irminger Sea cruise data. The black contour lines indicate saturation degree with respect to 2015 atmospheric concentrations. The white line shows winter mixed layer depth. The vertical lines show longitude bounds for the Irminger Sea region. The black dots show sampling depths. The location of the transect is shown in the map.

region south of Greenland could possibly be advected into the Irminger Sea on shorter timescales, but propagation rates are difficult to determine. In winter 2012–2013, convection was shallow and there was little dense water production in the entire subpolar gyre (Fig. 2b; Supplementary Fig. 1). During the subsequent winter of 2013–2014, dense water masses were produced by deep convection in the Labrador Sea as well as south of Greenland. However, in order for these water masses to potentially precondition the water column in the Irminger Sea for convection in winter 2014–2015, a fast eastward advection would be required. This is implausible given the observed mid-depth large-scale circulation[30]. The February to April mixed layer density evolution also reveals no eastward propagation of dense waters, particularly in 2013–2014. It appears instead that the deep mixed layers in the Irminger Sea arose locally in 2014–2015 by a gradual densification. That winter, deep convection resulted in the production of almost equally dense water masses in the Labrador Sea and the Irminger Sea. Therefore, advection of convectively formed water masses from outside the Irminger Sea cannot explain our observations. Hence, other mechanisms are responsible for the strong local convection.

**Atmospheric forcing and water column stratification.** The leading mode of atmospheric variability over the mid-latitude North Atlantic is the NAO. The positive phase of the NAO is characterized by a deeper Icelandic Low and stronger westerlies across the North Atlantic[31]. The NAO index (Supplementary Fig. 2) attained the highest value observed since the mid-1990s in winter 2014–2015. Atmospheric conditions during the positive phase of the NAO are conducive for the formation of westerly tip jets[32]. Tip jets are intense, periodic westerly winds that develop over the Irminger Sea as a result of the interaction of passing

extra-tropical cyclones with the high topography of southern Greenland[33,34]. These local wind phenomena are typically associated with high wind speeds and elevated sea-air heat fluxes over the Irminger Sea[19]. Due to their small spatial scale, coarse-resolved global climate models fail to simulate the magnitude of tip jet events around Greenland[32,35]. Consistent with the high NAO index, the number of westerly tip jet events as well as the winter mean total oceanic heat loss over the Irminger Sea during the winter of 2014–2015 attained values not seen since the mid-1990s (Fig. 3a; Supplementary Fig. 2).

As a measure of water column stratification we use the buoyancy frequency N, which represents the frequency at which a neutrally buoyant parcel will oscillate in a stably stratified fluid. It is calculated as:

$$N^2 = -\frac{g}{\rho_0}\frac{d\rho}{dz}, \qquad (1)$$

with g being the downward acceleration due to gravity and $\rho_0$ the mean density. A preconditioned water column with reduced stratification, for example through the presence of weakly stratified water at intermediate levels, facilitates deep convection before the onset of the convective season[13]. The temporal evolution of buoyancy frequency with depth based on Argo data from 2002 to 2015 illustrates the interannual variability of preconditioning in the Irminger Sea (Fig. 3b). There is a seasonal cycle of surface stratification connected to local winter convection. The lowest values of buoyancy frequency are observed in the winter mixed layers as a consequence of buoyancy loss from the surface, followed by a period of restratification of the upper 500–1,000 m in spring and summer. The deepest winter mixed layers (white bars) were observed in winters 2007–2008, 2011–2012 and 2014–2015. Compared with the moderate winters before 2007, buoyancy

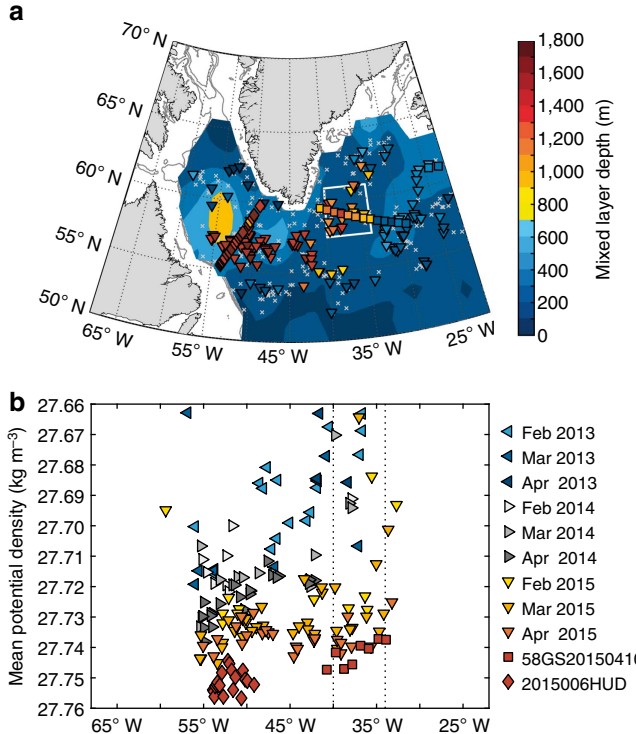

**Figure 2 | Wintertime mixed layer depth distribution.** February–April mixed layer depths greater than 80% of the deepest recorded mixed layer depth for individual Argo floats (triangles), Irminger Sea cruise data (58GS20150410) collected in April (squares) and Labrador Sea cruise data (2015006HUD) collecected in May (diamonds). Note that mixed layers for May Labrador Sea cruise data were isolated from the sea surface. Mixed layer depths were calculated using a semi-subjective method following Pickart et al.[22]. (**a**) Winter 2014–2015 data with 2000–2007 average winter mixed layer depths as contours[20]. The crosses show locations of all data points. The white box shows the region in the Irminger Sea selected for further analysis. (**b**) Mean potential density anomaly ($\sigma_0$) over the winter mixed layer of the February–April Argo data for 2013 (blue scale), 2014 (grey scale) and 2015 (red scale). The vertical lines show longitude bounds for the Irminger Sea region.

frequency at depth has generally been lower after the deep convection in winter 2007–2008 (ref. 20).

The impact of these deep convection events on water column stability becomes more evident by analyzing the mean buoyancy frequency from individual Argo profiles (Fig. 3c). While the features are robust over a wide depth range, the mean buoyancy frequency between 500 and 1,000 m is presented here. During the convective season (February–April, grey bars) in 2008, 2012 and 2015, the mean buoyancy frequency at depth was low as a consequence of convection. Deep convection during these three winters interrupted periods of restratification that led to higher values of mean buoyancy frequency at depth, most likely due to lateral advection or mixing of more stratified water masses into the region. This variability can partly be explained by trends deduced from least-squared fits over the low-passed envelope of mean buoyancy. The envelope was computed by considering every local minimum of the mean buoyancy frequency time series and smoothing the result using a running mean filter for the periods 2002–2007, 2008–2011 and 2012–2015. In general, the Irminger Sea has shifted towards a state of weaker mid-depth stratification after the strong convection in 2007–2008.

Deep convection occurs in the Irminger Sea despite strong initial stratification during severe winters in terms of heat

removal from the ocean (for example, winter 2007–2008 (ref. 20)). Although surface buoyancy loss in winter 2013–2014 led to deep mixing in the Labrador Sea and parts of the subpolar gyre, there is, as noted above, no indication that convectively-formed water masses from that winter advected into the Irminger Basin (Fig. 2b). The buoyancy frequency (Fig. 3b) substantiates this. The water column in fall 2014 reveals no particular sign of preconditioning for deep convection in 2014–2015. Combined, all these observations indicate that the strong winter surface buoyancy loss was the primary agent inducing deep convection in the Irminger Sea in winter 2014–2015, while preconditioning of the water column was only of secondary importance.

## Discussion

In the early twentieth century, deep convection was thought to occur in the Irminger Sea[36], but in the decades that followed, the Labrador Sea became the main area of interest for subsequent convection studies as it is the primary location for deep water formation in the North Atlantic. The Irminger Sea has, however, been re-identified as a region where deep convection takes place under favourable oceanic and atmospheric conditions[15–21]. The convection in winter 2014–2015 was the third deep water formation event in the Irminger Sea since the winter 2007–2008, with substantial impact on not only ocean circulation and stratification processes but also, as presented here, ocean biogeochemistry and carbon cycle variability.

The effect of convection and its variability in the Irminger Sea on ocean oxygen and anthropogenic carbon is illustrated by comparing saturation profiles at representative stations in April 2015 with cruise data from 1997 (ref. 37) and 2003 (ref. 38) (Fig. 4a,b). The saturation degree of oxygen and anthropogenic carbon in the Irminger Sea has undergone significant changes related to deep water formation variability. The formation of well-ventilated LSW during the mid-1990s is reflected by the elevated oxygen and anthropogenic carbon saturation values between 1,000 and 1,750 m depth in the 1997 profiles (blue curves). In 2003 (black curves) the saturation levels were reduced throughout most of the water column, a consequence of no deep ventilation between 1997 and 2003 and aging of the remnant water masses[6]. In contrast, elevated oxygen and anthropogenic carbon concentrations in 2015 in the Irminger Sea are a result of high convective activity. At the tail end of the convective season in 2015, the mixed layer was nearly saturated in oxygen and anthropogenic carbon (>90%) to a depth of 1,250 m.

The efficiency of the air-to-sea flux ultimately determines the degree of saturation and preformed concentrations of atmospheric gases in seawater. Preformed concentrations are set at the time of water mass formation, that is, the last time a water parcel is in direct contact with the atmosphere before subduction. For oxygen, preformed values are assumed to be close to saturation, however, the exact concentration can only be determined if measured directly. We observed a mean oxygen undersaturation of $3.6 \pm 0.9\%$, possibly driven by solubility effects due to the strong heat loss over winter and by the entrainment of old, less recently ventilated water masses[28]. Since oxygen concentrations at depth reflect the balance between supply and consumption, this needs to be taken into account if apparent oxygen utilization is used as in Feely[2] as a measure of biological activity, to avoid overestimation of respiration processes. The saturation degree of $SF_6$ in the winter mixed layer varied from undersaturated to supersaturated conditions with supersaturated values occuring at the base of the mixed layer (Fig. 1a). The consistency of this signal across stations indicates that this is not a measurement artefact. Supersaturation of $SF_6$ is

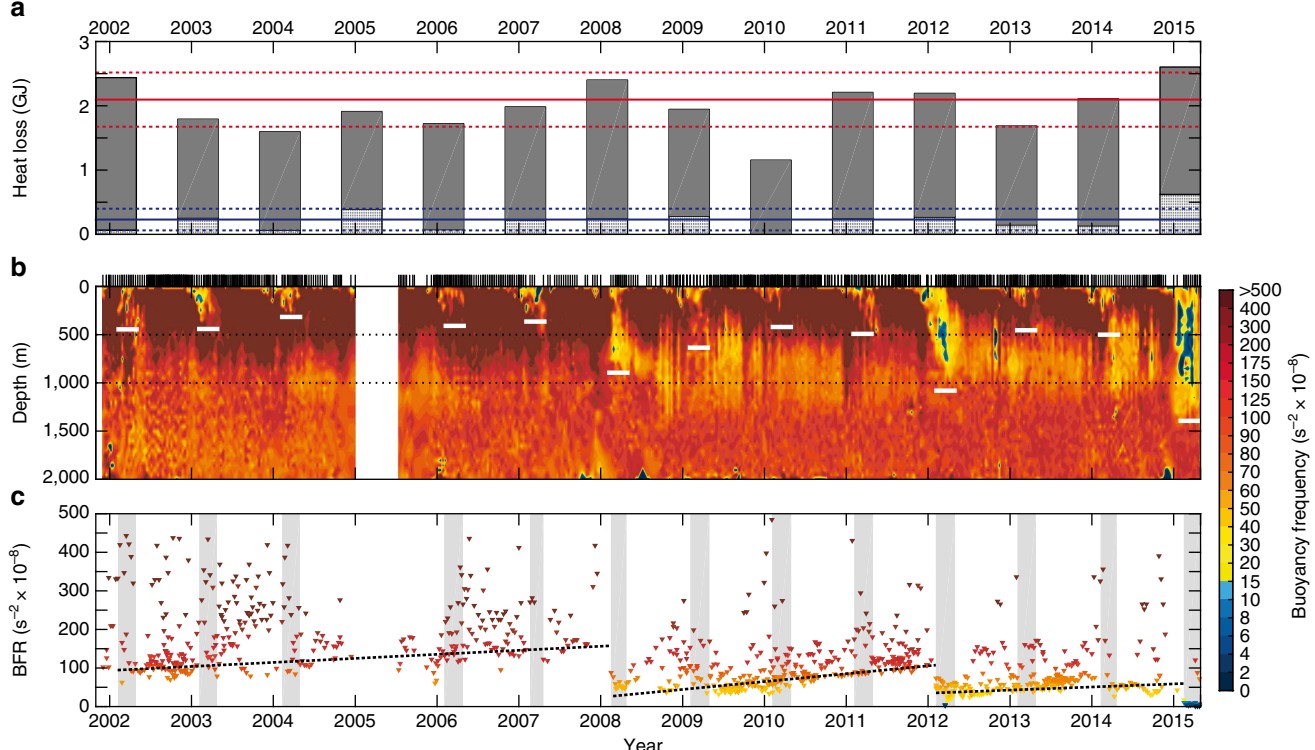

**Figure 3 | Temporal evolution of ocean heat loss, buoyancy frequency and mean buoyancy frequency in the Irminger Sea. (a)** The bars represent the integrated ocean heat loss from November to April based on ERA-Interim reanalysis data. The hatched part indicates the heat loss associated with tip jet events. The red solid line is the mean ocean heat loss from November to April from 1979 to 2015, the red dashed lines represent ±1 s.d. The blue solid line is the mean ocean heat loss associated with tip jet events from November to April from 1979 to 2015, the blue dashed lines represent ±1 s.d. **(b)** Interpolated buoyancy frequency with depth and time, based on Argo data. The white bars represent the 95th percentile of the February–April mixed layer depth (greater than 80% of deepest recorded mixed layer depth of individual floats). The black dotted lines highlight 500 and 1,000 m depth. The vertical bars on top denote the sampling time of each profile in the Irminger Sea. **(c)** Mean buoyancy frequency over 500–1,000 m. The grey bars highlight the convective season (February–April). The black dotted lines are least-square fits over the low-passed envelope of mean buoyancy frequency for 2002–2007 ($R^2 = 0.50$), 2008–2011 ($R^2 = 0.65$) and 2012–2015 ($R^2 = 0.20$).

induced by bubble formation at high wind speeds[25]. The reason why this water is found at the base of the mixed layer is likely because the highest winds speeds also result in the strongest heat loss and most active convection. Our measurements thus show that transient tracers such as $SF_6$ are not necessarily undersaturated at the time of water mass formation[24], and that physical processes such as bubble-mediated gas transfer may influence saturation values during winter convection.

Changes in the inventory of anthropogenic carbon reflect the interannual variability of the convective activity in the subpolar gyre (Fig. 4c). Exploiting the almost exponential time history of atmospheric $CO_2$ and evoking the theory of transient steady-state[39], the anthropogenic carbon concentrations in the ocean are expected to increase by 1.90% per year over our considered time period. Therefore, the anthropogenic carbon inventory is expected to increase by $1.20 \pm 0.16$ mol C m$^{-2}$ per year in the absence of circulation changes and only based on rising atmospheric $CO_2$ concentrations. In contrast, between 1997 and 2003, anthropogenic carbon inventories increased by $0.63 \pm 0.50$ mol C m$^{-2}$ per year, which leads to a lower-than-expected inventory in 2003 due to absence of ventilation processes and export of LSW from the western subpolar North Atlantic. Between 2003 and 2015 however, the uptake rate of anthropogenic carbon was $1.94 \pm 0.29$ mol C m$^{-2}$ per year and clearly larger than expected from the atmospheric $CO_2$ increase alone. The fact that ocean variability increased the anthropogenic

carbon inventory rate by a factor three between 1997 and 2015, indicates that frequent observations of physical and biogeochemical parameters in the subpolar North Atlantic are required to fully quantify anthropogenic carbon uptake variability and to distinguish between trends associated with long-term climate change and signals attributed to natural variability.

Physical properties have been monitored for many decades in the Atlantic, covering several phase shifts and amplitude changes of modes of atmospheric variability such as the NAO. Carbon chemistry data on the other hand have only been systematically collected on a repeat basis since the early 1990s. As the time span of the record is extended, the strong response of ventilation and anthropogenic carbon storage in the Atlantic to variability in atmospheric forcing becomes more evident[5,8,9]. In particular, the observations presented here reveal a strong link between oceanic heat loss enhanced by numerous tip jet events, ventilation and anthropogenic carbon storage in the Irminger Sea, which is one of the deep water formation areas in the North Atlantic. On a broader scale, the variations in ventilation that are being uncovered and attributed with our growing observational data base may also help to understand the decadal variability clearly present in North Atlantic sedimentary records[40,41]. For a robust prediction of future changes in convective processes in the subpolar gyre and their impacts on ocean oxygen and anthropogenic carbon, atmospheric forcing needs to be well-represented in global climate models by resolving small-scale atmospheric patterns such as the Greenland tip jet.

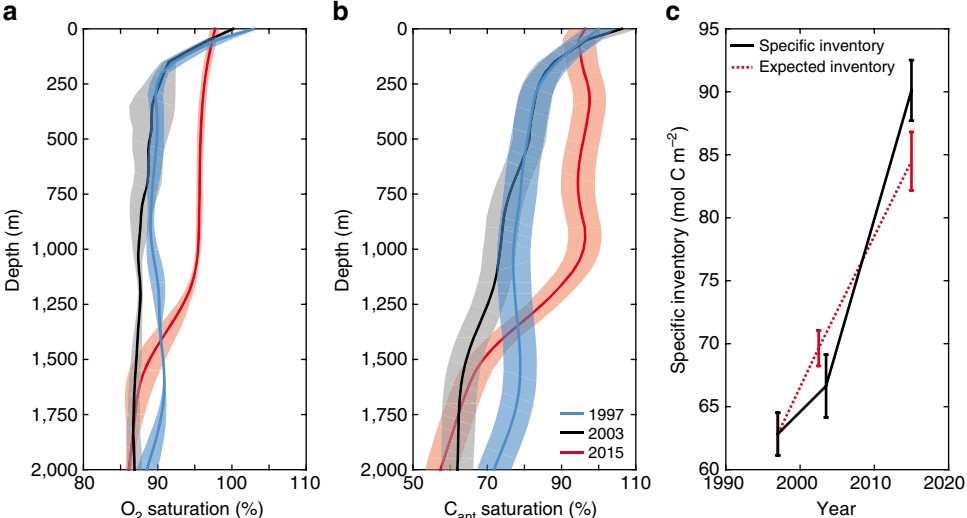

**Figure 4 | Water column properties between 0 and 2,000 m for three selected cruises in the Irminger Sea.** (**a**) Oxygen and (**b**) anthropogenic carbon mean saturation with depth for selected stations for 1997 (blue curves), 2003 (black curves) and 2014 (red curves) in the Irminger Sea based on cruise data (06MT19970815, 06MT20030723 and 58GS20150410). Note the different scales in (**a,b**). (**c**) The specific anthropogenic carbon inventory (black) and the expected evolution of anthropogenic carbon inventory based on actual atmospheric $CO_2$ increase and 1997 conditions (red) over 100–2,000 m in the Irminger Sea. The shading and the error bars represent ±1 s.d. over the station data, for (**b,c**) including the uncertainty of the TTD method.

## Methods

**Data.** The Argo program consists of freely drifting profiling floats, which provide real-time salinity and temperature measurements of the upper 2,000 m of the ocean with accuracies of ±0.01 for salinity and ±0.002 °C for temperature. We use 22,764 Argo profiles obtained in the subpolar gyre from 2002 to 2015 with a quality flag of good (1) or probably good (2). Cruise data was collected during the cruises 58GS20150410, crossing the Irminger Sea from 16–20 April 2015, and 2015006HUD, crossing the Labrador Sea from 2–24 May 2015. The location of the stations, used for Fig. 4, for the Meteor cruises in 1997 (06MT19970815, collected from 15 August to 9 September)[37] and 2003 (06MT20030723, collected from 23 July to 19 August)[38] and 58GS20150410 cruise data are presented in Supplementary Fig. 3.

At 58GS20150410, direct measurements of pressure, conductivity, temperature and dissolved oxygen were obtained with a Seabird 911+ CTD. The CTD measurements were calibrated against *in situ* samples obtained at all Niskin bottle sample depths (usually 12) at every cast, following the Global Ocean Ship-based Hydrographic Investigations Program (GOSHIP) calibration procedure[42]. Bottle salinities were analysed with a salinometer with an accuracy of ±0.003. Winkler titration was performed to analyse oxygen concentration of water samples using a potassium iodate solution as a standard. The precision was ±0.005 ml l$^{-1}$. CFC-12 and $SF_6$ was measured with a similar purge and trap system as described in Bullister and Wisegarver[43] and Stoven and Tanhua[44], with gas chromatographic separation and electron capture detection. The precision was ±1.3% for CFC-12 concentrations and, due to some analytical issues, ±6.2% for $SF_6$ concentrations. Solubilities of CFCs and $SF_6$ are estimated from salinity and temperature relations according to Warner and Weiss[45] and Bullister and Wisegarver[46]. The accuracy of the atmospheric record for both CFC-12 and $SF_6$ is much better than 1%, hence negligible. The accuracy of the solubility functions of CFC-12 and $SF_6$ are about 1.5% and 2%, respectively. The combined calibration and analytical uncertainties for CFC-12 saturation values is therefore 2.8% and for $SF_6$ 8.2%. According to Weiss[47], the accuracy for the oxygen solubility function in seawater is ±0.015 ml l$^{-1}$, therefore the combined uncertainty for oxygen saturation values is in the order of 0.3%.

**Mixed layer depth estimates.** Mixed layer depths were estimated by manual inspection of individual profiles. Following Pickart *et al.*[22], the upper and lower depth limits of the mixed layer were first determined visually; and over that depth range, the mean density and its standard deviation were estimated. If no estimate could be made from the density profile, the temperature or salinity profile was used instead. The intersections between the original profile and the two-standard deviation envelope was defined as the mixed layer depth (Supplementary Fig. 4). This criterion also enabled determination of the depths of mixed layers isolated from the surface (Supplementary Fig. 5), which have been shown to occur in the subpolar gyre, either in the form of stacked multiple mixed layers or during the early phase of surface restratification[19,22]. For consistency with previous studies[20], only mixed layers deeper than 80% of the maximum recorded mixed layer depth

for each winter and each float were included in the analysis to avoid a shallow bias associated with the non-uniform spatial and temporal character of convection.

**Anthropogenic carbon.** The concentration of anthropogenic carbon ($C_{ant}$) was estimated with the TTD method[26] based on CFC-12 data and $SF_6$ data. The concentration of any passive, inert tracer is determined at any place $x$ at any time $t$ by the source function of the tracer at sea surface $C^0(t)$ and the transit time distribution $\mathcal{G}(x, \tau)$:

$$C(x, t) = \int_0^\infty C^0(t-\tau)\mathcal{G}(x, \tau)d\tau' \qquad (2)$$

This method allows to establish a transfer function between measured tracer concentration and $C_{ant}$ with three assumptions made: (1) increasing $C_{ant}$ concentration in the ocean does not affect the biological uptake of carbon, (2) the ocean circulation is in steady-state and (3) $C^0(t)$ has no spatial dependence for a given $x$ (ref. 26).

TTDs can be approximated by inverse Gaussian functions at each location in the interior ocean:

$$\mathcal{G}(\tau, \Gamma, \Delta) = \sqrt{\frac{\Gamma^3}{4\pi\Delta^2\tau^3}}\, exp\left(\frac{-\Gamma(\tau-\Gamma)^2}{4\Delta^2\tau}\right), \qquad (3)$$

where $\Gamma$ is the mean transit time and $\Delta$ defines the width of the TTD. In the subpolar gyre, the ratio $\Gamma/\Delta$ is assumed to be 1 (ref. 6). The source function depends on the atmospheric history (compiled by Bullister[48] for CFCs and $SF_6$; Mauna Loa updated records for $CO_2$ (ref. 49)) and the tracer solubility in sea water. Since the atmospheric history of CFCs and $SF_6$ is well documented, the TTD at any place $x$ can be defined with direct observations of CFC-12 or $SF_6$, respectively. In case of excess $SF_6$, that is, supersaturated values within the mixed layers, age estimates are set to zero (ref. 24). Observed temperature ($T$) and salinity ($S$) are used to estimate preformed alkalinity ($Alk^0$)[50]. We use thermodynamic equations of the seawater $CO_2$ system[51] and the $CO_2$ dissociation constants of (ref. 52) refitted by (ref. 53) to calculate the time history in the surface mixed layer for anthropogenic $CO_2$:

$$C_{ant,0}(t) = C_{eq}\left(T, S, Alk^0, pCO_2(t)\right) - C_{eq}\left(T, S, Alk^0, pCO_2 = 280\,p.p.m.\right) \quad (4)$$

Here, $C_{ant,0}$ is the difference between the total inorganic carbon $C_{eq}$ at air-sea equilibrium with respect to the atmospheric $CO_2$ concentration at time $t$ and at preindustrial atmospheric $CO_2$ levels (280 p.p.m.). The saturation of $C_{ant}$ in the ocean is the ratio between actual $C_{ant}$ concentration and $C_{ant,0}$.

$C_{ant}$ inventories are estimated by integrating $C_{ant}$ concentrations over 100–2,000 m: $INV = \int_0^z C_{ant}\, \varrho\, dz$. Inventory increase rates are the difference in column inventory divided by the time in years between the measurements. The method-based uncertainty for $C_{ant}$ is ±6 µmol kg$^{-1}$ (ref. 26). For the inventory estimates, uncertainties were calculated by randomly propagating this standard error over depth for the selected stations which led to an uncertainty of ±1 mol C m$^{-2}$. The expected rise of $C_{ant}$ is 1.9% a$^{-1}$, calculated for equilibrium

conditions in 1997 using the thermodynamic equations of the seawater $CO_2$ system and assuming an exponential time history of dissolved $CO_2$ in the surface mixed layer[6]. Multiplying this expected increase with the $C_{ant}$ concentration in 1997 yields the $C_{ant}$ concentration increase expected from rising atmospheric $CO_2$ levels only.

**Atmospheric forcing.** The atmospheric circulation is analysed using the ERA-Interim reanalysis data product, archived by the European Centre for Medium Range Weather Forecasts (ECMWF) and available at http://apps.ecmwf.int/datasets/. The NAO, a large-scale pressure oscillation with centres of action near Iceland and the Azores is the leading mode of variability in the North Atlantic region[54]. An index of the NAO can be defined using the sea-level pressure at Stykkisholmur, Iceland and Lisbon, Portugal[54]. The NAO is said to be in its positive state when the sea-level pressure near Iceland is anomalously low, while that near the Azores is anomalously high. Here, we used the sea-level pressure from the ERA-Interim to define the NAO Index. The heat loss associated with tip jet events is the heat loss during a fixed time period of 24 h, that is, ± 12 h from the peak wind speed during each event.

**Data availability.** The data that support the findings of this study are available from the corresponding authors (F.F. and A.O., are.olsen@gfi.uib.no) upon request and are available within the article.

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

## Acknowledgements

The authors would like to thank the captain of G.O. SARS, John H. Johnsen, and his crew, and the scientists that took part in the cruise to the Irminger Sea, especially Kristin Jackson, Jörg Schwinger and Ailin Brakstad. Ulysses Ninnemann provided initial input on the paper. The European Centre for Medium-Range Weather Forecasts is acknowledged for access to the the ERA-Interim reanalysis. The Argo Program (part of the Global Ocean Observing System) is acknowledged for collecting data and making these freely available (http://www.argo.ucsd.edu). The 2016 AR7W data used in this work were collected in the 2016-006 cruise of the CCGS Hudson (2015006HUD) performed as a part of the Atlantic Zone Off-Shelf Monitoring Program (AZOMP) by Fisheries and Oceans Canada. The Meteor cruise data were retrieved from GLODAPv2. A.O and F.F appreciate funding from the SNACS project (229752), that is part of the KLIMAFORSK program of the Norwegian Research Council. E.J. and B.R. received funding from the NRC project VENTILATE (229791). Support for this work was provided by the European Union 7th Framework Programme (FP7 2007–2013) under Grant agreement no. 308299 NACLIM Project (K.V.). K.M. received funding from the Natural Sciences and Engineering Research Council of Canada. This is a contribution to the BIGCHANGE project of the Bjerknes Center for Climate Research.

## Author contributions

F.F., A.O., E.J. and B.R. collected the 58GS20150410 data and I.Y. provided the 20152006HUD data. F.F. carried out all data analysis, apart from the atmospheric forcing, which was done by K.M. All authors jointly interpreted the results in this paper, which was mainly written by F.F. with contributions from A.O., K.V., I.Y. and K.M.

## Additional information

**Competing financial interests:** The authors declare no competing financial interests.

