## [Peer review file · Nature Communications]

Reviewers' comments:

Reviewer #1 (Remarks to the Author):

A. Summary of the key results

This paper describes recent, and deep convection in the Irminger Sea, directly observed from hydrographic section data (rare) and also from Argo float data (which provides a multi-year context). The deep mixing is anomalous for its magnitude (>1000m) and location (east of Greenland and south of Iceland). The observations are novel as they present tracer measurements (oxygen, SF₆, carbon) as well as hydrographic data during convection, and show the degree of saturation during deep mixing of oxygen and SF₆, which is useful for downstream tracer studies, e.g. in the subtropical Atlantic where tracer measurements are used to estimate time since ventilation, which assumes certain levels of saturation during ventilation.

The authors show that this deep convection is primarily associated with wintertime conditions in the 2014-15 winter, which is important to place the tracer measurements in context, and also to identify drivers of convection.

The manuscript is greatly improved from the original submission.

B. Originality and interest: if not novel, please give references

To my knowledge these are the first observations of SF₆ during deep convection, and the observations of deep convection in an unusual location with supporting chemistry is novel.

C. Data & methodology: validity of approach, quality of data, quality of presentation

* It would be useful if the authors could relate their supersaturation of SF₆ in the manuscript to the precision issues mentioned in the methods. While the precision may not influence the pattern found (higher saturation at the base of the mixed layer), could it change the anomaly from being supersaturated to saturated? How do the precision estimates in percentages relate to the saturation levels in percentages? I am assuming (but the text does not specify) that the reported precision for CFC-12 and SF₆ is a concentration rather than a saturation. Please clarify.

* The cruise identifiers (58GS20150410) are included in the figure caption (Fig. 1) but are only identified in the supplementary as being the Irminger cruise in April. Are there really 1600m actively mixing layers in the Labrador Sea in May? This seems late. The method for estimating the mixed layer depth isn't fully clear from the manuscript (L375-397) or supplementary (Fig SI2). In particular, the surface 500m are not shown in Fig SI2, and the methods mentions "this criterion also enabled determination of the depths of mixed layers isolated from the surface". How are the mixed layers identified via this method distinguishable from recently restratified water? From what I can glean of the method, the mean density and standard deviation are estimated over the mixed layer, but is this from the surface to mixed layer depth or from 500 to the mixed layer depth (the latter being suggested by the axes limits in Fig. SI2).

The authors cite references 19, 24 with mixed layers isolated from the surface, but unless Frob et al. have further information, I don't think the mixed layers described here can distinguish between actively mixing or recently restratified waters. Please clarify the method, and exercise some caution when describing these mixed layers as "isolated from the atmosphere" vs restratified. The

results will stand either way.

D. Appropriate use of statistics and treatment of uncertainties

* What method is used to determine statistical significance (or lack thereof) on the trends in Fig. 3c? Some detail would be useful as the caption specifies a "low-passed envelope of integrated buoyancy frequency", though the low-pass is not described anywhere.

E. Conclusions: robustness, validity, reliability

* I think the main messages here are that (1) the Irminger Sea cannot be neglected when considering deep water formation; (2) that profiles of tracers during convection show high levels of saturation throughout the top 1000m (with the alternative being that the convective processes would be started and finished too quickly for such a large volume of water to be completely ventilated/brought into equilibrium with the atmosphere). The speculation regarding SF6 supersaturation is interesting. I would suggest adjusting L241 "influence saturation values.." to "may influence saturation values..".

Conclusion (1) is used to highlight the importance of small scale atmospheric events (tip jets) though this isn't described directly.

F. Suggested improvements: experiments, data for possible revision

* In Fig. 3c, for the integrated buoyancy frequency, how are the depth limits of 750 to 1000m chosen? Do the results change substantially if limits of 500 to 1000 are used? If BFR (N2) is a useful measure for stratification, why do you then integrate again before using it in Fig. 3c to discuss tendencies in convection/restratification?

* When were the data in Fig. 4 collected (what time of year?)

G. References: appropriate credit to previous work?

Yes

H. Clarity and context: lucidity of abstract/summary, appropriateness of abstract, introduction and conclusions

Yes, these are greatly improved and place the focus of the paper closer to what can be discussed from the analysis presented. These observations are remarkable, and the manuscript highlights them well.

Reviewer #3 (Remarks to the Author):

Review of: "Irminger Sea deep convection injects oxygen and anthropogenic carbon to the ocean interior"

This paper presents observations from the subpolar North Atlantic, focusing on deep convection during the 2014-2015 winter. In principle the observations presented are interesting and merit publication, however, there are several elements of the paper that are difficult to follow due to poor presentation. I don't think the paper is suitable for publication in its present form due to these issues.

Abstract

In 21-22: "[T]ip jet events" is jargon and should be replaced with a phrase that people not familiar with the phenomenon can understand.

In 24-25: "[A]lmost tripled the anthropogenic carbon storage" of what region? The whole ocean? The subpolar N. Atlantic?

Introduction

In 37: I don't think "subpolar" should be capitalized---but if it should then it should be consistent throughout the paper (e.g., In 54).

In 47: The phrase, "acceleration in carbon storage" doesn't make sense. Carbon storage might increase (perhaps at an increasing rate), but I don't see how storage can accelerate: storage is not a rate.

Results

In 78-89: It is unclear to me from this description how the authors are using SF₆ saturation values to determine that strong local convection has happened. This sentence, "The SF₆ saturation values vary between 86 and 125 %, which given the typical saturations of 86 % that are observed in the ocean surface, testify to the recent formation through convective processes," is particularly unhelpful. Does relatively high supersaturation indicate convection or relatively low undersaturation? What is meant by "recent formation?" Recent water mass formation? This section is not acceptable as written.

In 90-94: The authors have not provided sufficient information (including within the supplemental text) to understand how Cant has been computed. What are the assumptions used to obtain C₀? The preceding text just illustrated that surface concentrations of SF₆ are not at equilibrium.

In 105-106: The percent-saturation of anthropogenic carbon needs to be defined.

Fig 2b: This figure is exceptionally ineffective (bad) at conveying the intended point. I would think the notion that 2014-2015 had unusually deep mixed layers relatively easy to show, but the authors have chosen to obfuscate this message by presenting a cloud of overlapping dots that must be matched by color and shape a poorly labeled legend. Perhaps panel a could be relied upon to convey a sense of the spatial structure in the anomalies and panel b could focus on the time-evolution seasonally, including climatological variations and the year of interest.

In 145: What are "westerly tip jets?" The following sentence says where they develop, but not what they are.

Discussion

In 209-211: This sentence, "In only a short period of time, saturation of oxygen and anthropogenic carbon in the Irminger Sea has undergone significant changes related to deep water formation variability," appears to suggest a secular trend: are we thinking that this is a secular trend or interannual to decadal scale variability?

In 228-231: What is the proposal here? The notion that preformed O₂ is not always at equilibrium is hardly new. The problem is that better estimate of AOU would require knowing this disequilibrium component---which is highly variable and thus not trivial.

In 242: I would change "storage rate" to "rate of accumulation".

In 247: How much of the increased rate of accumulation can be attributed to rising atmospheric CO₂ versus interannual variation in circulation?

Reviewer 1

A. Summary of the key results

This paper describes recent, and deep convection in the Irminger Sea, directly observed from hydrographic section data (rare) and also from Argo float data (which provides a multi-year context). The deep mixing is anomalous for its magnitude (>1000m) and location (east of Greenland and south of Iceland). The observations are novel as they present tracer measurements (oxygen, SF₆, carbon) as well as hydrographic data during convection, and show the degree of saturation during deep mixing of oxygen and SF₆, which is useful for downstream tracer studies, e.g. in the subtropical Atlantic where tracer measurements are used to estimate time since ventilation, which assumes certain levels of saturation during ventilation.

The authors show that this deep convection is primarily associated with wintertime conditions in the 2014-15 winter, which is important to place the tracer measurements in context, and also to identify drivers of convection.

The manuscript is greatly improved from the original submission.

B. Originality and interest: if not novel, please give references

To my knowledge these are the first observations of SF₆ during deep convection, and the observations of deep convection in an unusual location with supporting chemistry is novel.

C. Data & methodology: validity of approach, quality of data, quality of presentation

* It would be useful if the authors could relate their supersaturation of SF₆ in the manuscript to the precision issues mentioned in the methods. While the precision may not influence the pattern found (higher saturation at the base of the mixed layer), could it change the anomaly from being supersaturated to saturated? How to the precision estimates in percentages relate to the saturation levels in percentages? I am assuming (but the text does not specify) that the reported precision for CFC-12 and SF₆ is a concentration rather than a saturation. Please clarify.

We thank the reviewer for this comment. The precision in the supplementary material is indeed referred to the measured concentration of CFC-12 and SF₆, respectively. We added a paragraph to describe the accuracy of the solubility function and the resulting total precision. The precision for the solubility function is $\pm 1.5\%$ for CFC-12 (Warner et al., 1985) and $\pm 2\%$ for SF₆ (Bullister et al., 2002). The accuracy of the atmospheric record for both CFC-12 and SF₆ is smaller than 1%, hence negligible. The total uncertainty for saturated values is the sum of the relative errors, hence for CFC-12 $\pm(1.5\% + 1.3\%) = \pm 2.8\%$ and for SF₆ $\pm(2\% + 6.2\%) = \pm 8.2\%$, where 1.3% and 6.2% are the uncertainties in the measured concentrations. The mean saturation for SF₆ at the base of the mixed layer for the 7 westernmost stations (the stations, where the supersaturated values at the base of the mixed layer are detected) is $116.2 \pm 8.2\%$. Therefore, within the range of uncertainty, supersaturated values are observed. We revised this section, also due to the comments made by reviewer 3 and added these uncertainties. Further, in the supplementary material we added a description of how these uncertainties were calculated .

* The cruise identifiers (58GS20150410) are included in the figure caption (Fig. 1) but are only identified in the supplementary as being the Irminger cruise in April. Are there really 1600m actively mixing layers in the Labrador Sea in May? This seems late. The method for estimating the mixed layer depth isn't fully clear from the manuscript (L375-397) or supplementary (Fig SI2). In particular, the surface 500m are not shown in Fig SI2, and the methods mentions "this criterion also enabled determination of the depths of mixed layers isolated from the surface". How are the mixed layers identified via this method distinguishable from recently restratified water? From what I can glean of the method, the mean density and standard deviation are estimated over the mixed layer, but is this from the surface to mixed layer depth or from 500 to the mixed layer depth (the latter being suggested by the axes limits in Fig. SI2).

We appreciate this comment and revised the method section in order to clarify the method used to estimate the mixed layer depth. First, the data collected during the cruise crossing the Labrador Sea in May, 2015 (expocode: 2015006HUD) is only presented in Figure 2. Indeed, the presented mixed layers from this cruise are not active, but detached from the surface, probably due to restratification in late spring and early summer. We chose to plot these data in order to illustrate the deep convection in the Labrador Sea that particular winter. We clarified this in the manuscript.

The determination of the mixed layer depth is a semi-subjective process following Pickart et al. (2002). Each profile (CTD or Argo) is analysed individually. First, the extent of the mixed layer is estimated visually, that means, the upper and lower depth limit of the mixed layer is determined. The mean and standard deviation over this depth range is calculated. The point where the two-standard deviation envelope crosses the density profiles is determined as the depth of the mixed layer. The use of this method is particularly helpful for noisy profiles, where automatic evaluations of mixed layer depths fail. In case of doubt, the mixed layer depth was set to 'missing value'. We revised the Figure SI2 accordingly and added an example profile from the Labrador Sea cruise (Figure SI3) to illustrate a mixed layer that is isolated from the surface.

The authors cite references 19, 24 with mixed layers isolated from the surface, but unless Frob et al. have further information, I don't think the mixed layers described here can distinguish between actively mixing or recently restratified waters. Please clarify the method, and exercise some caution when describing these mixed layers as "isolated from the atmosphere" vs restratified. The results will stand either way.

Thank you for that remark. Indeed, we do not distinguish between mixed layers that are active and those that are isolated from the atmosphere (either because there are stacked multiple mixed layers or due to restratification of the surface). We added "This criterion also enabled determination of the depths of mixed layers isolated from the surface (Figure SI 3, supplementary material), which have been shown to occur in the subpolar gyre either in the form of stacked multiple mixed layers or during the beginning phase of surface restratification [24, 19]." to make this more clear in the method section. We cite Våge et al. (2008) and Pickart et al. (2002) here, because the same method has been described and applied here.

D. Appropriate use of statistics and treatment of uncertainties

* What method is used to determine statistical significance (or lack thereof) on the trends in Fig. 3c? Some detail would be useful as the caption specifies a "low-passed envelope of integrated buoyancy frequency", though the low-pass is not described anywhere.

Thanks for noticing this. We calculate the local minima of mean buoyancy frequency data and use a running mean over each of the three time periods to smooth the data. Then we use a linear regression model to analyse the trends. The R squared value is shown for all the three periods. We have explained this more clearly in the revised manuscript.

E. Conclusions: robustness, validity, reliability

* I think the main messages here are that (1) the Irminger Sea cannot be neglected when considering deep water formation; (2) that profiles of tracers during convection show high levels of saturation throughout the top 1000m (with the alternative being that the convective processes would be started and finished too quickly for such a large volume of water to be completely ventilated/brought into equilibrium with the atmosphere). The speculation regarding SF6 supersaturation is interesting. I would suggest adjusting L241 "influence saturation values.." to "may influence saturation values..".

Conclusion (1) is used to highlight the importance of small scale atmospheric events (tip jets) though this isn't described directly.

We agree with the comments made above. Indeed, the role of the Irminger Sea cannot be neglected if deep convection in the subpolar gyre is discussed, especially in a situation as in winter 2014-15, when the mean density of the produced water masses is comparable to these in the Labrador Sea. We rephrased the section 'atmospheric forcing' also due to comments made by Reviewer 3 and describe tip jets more detailed and argue that due to their small spatial scale, they are under-represented by global models (e.g. Moore et al., 2003, DuVivier et al., 2016).

We agree, that the point made for SF6 is more speculative and rephrased the referred statement as suggested.

F. Suggested improvements: experiments, data for possible revision

* In Fig. 3c, for the integrated buoyancy frequency, how are the depth limits of 750 to 1000m chosen? Do the results change substantially if limits of 500 to 1000 are used? If BFR (N2) is a useful measure for stratification, why do you then integrate again before using it in Fig. 3c to discuss tendencies in convection/restratification?

In Figure 3c we want to show how the water column stability at depth has changed over time, in particular with respect to the deep convective events in 2007/08, 2011/12 and 2014/15. In our first version, which was submitted to Nature Geoscience we showed BFR anomalies (measured BFR - monthly mean BFR from 2002-2015) over 500-1000m. Reviewer 2 for that version argued that it might be better to show integrated BFR instead,

which we did for the revised manuscript. However, the comment above made by Reviewer 1, now led us to rethink this decision.

We looked at different products (mean BFR, integrated BFR and BFR anomaly) over several different depth levels. The results do not change. For all quantities, there is a positive trend in the first two periods, that is statistical significant. For the third period, there is no statistical significant trend.

In the end we decided to keep it simple and show the mean BFR over 500-1000m depth range in the new, revised manuscript.

* When were the data in Fig. 4 collected (what time of year?)

For the 1997 cruise, data was collected from 8/15 to 9/9/1997, for the 2003 cruise from 7/23 to 8/29/2003 and for the 2015 cruise from 4/10 to 4/26/2015. We added this information in the supplementary material. That means, the earlier two cruises are summer cruises with very shallow mixed layers., so that surface conditions (0-100m) might not be comparable for the selected cruises. We therefore decided to exclude the first 100m in our inventory calculations. We assume, below the first 100m, conditions are comparable for the selected cruises. We updated Figure 4a-c accordingly as well as the inventory rate calculations. The main results are not affected by this decision.

G. References: appropriate credit to previous work?

Yes

H. Clarity and context: lucidity of abstract/summary, appropriateness of abstract, introduction and conclusions

Yes, these are greatly improved and place the focus of the paper closer to what can be discussed from the analysis presented. These observations are remarkable, and the manuscript highlights them well.

Reviewer 3

This paper presents observations from the subpolar North Atlantic, focusing on deep convection during the 2014-2015 winter. In principle the observations presented are interesting and merit publication, however, there are several elements of the paper that are difficult to follow due to poor presentation. I don't think the paper is suitable for publication in its present form due to these issues.

Abstract

In 21-22: "[T]ip jet events" is jargon and should be replaced with a phrase that people not familiar with the phenomenon can understand.

Thank you for pointing this out. We revised the entire abstract, but we added a definition for tip jet events in the section “Atmospheric forcing and water column stratification” (see point below).

In 24-25: “[A]most tripled the anthropogenic carbon storage” of what region? The whole ocean? The subpolar N. Atlantic?

We added “[...] in the Irminger Sea” in order to clarify this statement.

Introduction

In 37: I don't think "subpolar" should be capitalized---but if it should then it should be consistent throughout the paper (e.g., In 54).

We agree and use 'subpolar' (non-capitalized) throughout the manuscript.

In 47: The phrase, "acceleration in carbon storage" doesn't make sense. Carbon storage might increase (perhaps at an increasing rate), but I don't see how storage can accelerate: storage is not a rate.

Thanks for pointing this out. We agree, therefore we rephrased this statement to: “A recent acceleration in anthropogenic carbon storage rates [...]”;

Results

In 78-89: It is unclear to me from this description how the authors are using SF6 saturation values to determine that strong local convection has happened. This sentence, "The SF6 saturation values vary between 86 and 125 %, which given the typical saturations of 86 % that are observed in the ocean surface, testify to the recent formation through convective processes," is particularly unhelpful. Does relatively high supersaturation indicate convection or relatively low undersaturation? What is meant by "recent formation?" Recent water mass formation? This section is not acceptable as written.

We appreciate this comment and we revised this paragraph, also with respect to the comments made by reviewer 1. Hopefully, it is more clear now. As pointed out by Tanhua et al. (2008), surface saturation values for SF6 of 86% are observed in the mid latitude North Atlantic and are in general undersaturated with respect to atmospheric values during water mass formation. However, we observe saturation values within the winter mixed layer that are mostly higher than 86%. Furthermore, we do observe supersaturated values of SF6 at the base of the mixed layer (mean value of $116.2 \pm 8.2\%$ for the first 7 stations to the west with that particular feature). High saturation values indicate recent contact to the atmosphere, hence are indicative for the recent, that is during winter 2014/15, formation of this water mass.

In 90-94: The authors have not provided sufficient information (including within the supplemental text) to understand how Cant has been computed. What are the assumptions used to obtain C0? The preceding text just illustrated that surface concentrations of SF6 are not at equilibrium.

We thank the reviewer for that comment. We revised the section “TTD method” in the supplementary material. We added the equation (3) that shows, how C0 for anthropogenic carbon is calculated. We also list the assumptions, the method makes in order to estimate anthropogenic carbon.

In 105-106: The percent-saturation of anthropogenic carbon needs to be defined.

We agree with the reviewer. We added in the method section the definition for the saturation of anthropogenic carbon. Basically, the ratio between measured anthropogenic carbon and the saturated anthropogenic carbon for atmospheric values in 2015 is defined as the degree of saturation of anthropogenic carbon.

Fig 2b: This figure is exceptionally ineffective (bad) at conveying the intended point. I would think the notion that 2014-2015 had unusually deep mixed layers relatively easy to show, but the authors have chosen to obfuscate this message by presenting a cloud of overlapping dots that must be matched by color and shape a poorly labeled legend. Perhaps panel a could be relied upon to convey a sense of the spatial structure in the anomalies and panel b could focus on the time-evolution seasonally, including climatological variations and the year of interest.

We agree, this Figure 2b is difficult and does not illustrate well enough the point that we tried to make. We wanted to show that there is no eastward propagation of winter mixed layer depth, hence convection took place locally. For clarification, we decided to plot the mean density over the mixed layer depth instead.

This now shows nicely that a) during one convective season there is a gradual densification with time, b) there is no eastward densification in time, meaning that there is no eastward advection under the considered time scale, c) there was little dense water production in winter 2012/13 in the entire SPG, dense water was produced in winter 2013/14 in the Labrador Sea and south of Greenland but the densest water was produced in winter 2014/15 in the entire SPG and d) in winter 2014/15 the water produced in the Labrador and Irminger Sea are of equal density.

In 145: What are "westerly tip jets?" The following sentence says where they develop, but not what they are.

We agree and rephrased the referred sentence: "[...] of westerly tip jet events. Tip jets are intense, periodic westerly winds that develop over the Irminger Sea as a result of the interaction of passing extra-tropical cyclones with the high topography of southern Greenland [32, 33]. These small-scale wind phenomena are typically associated with high wind speeds and elevated sea-air heat fluxes over the Irminger Sea." We hope, this is more clear now in the revised manuscript.

Discussion

In 209-211: This sentence, "In only a short period of time, saturation of oxygen and anthropogenic carbon in the Irminger Sea has undergone significant changes related to deep water formation variability," appears to suggest a secular trend: are we thinking that this is a secular trend or interannual to decadal scale variability?

No, this is most certainly no secular trend – a number of three cruises would not allow such a statement. With the following sentences we put the statement in context. We took out 'In only a short period of time' to clarify that we do not describe trends here, but rather describe the effect that deep water formation variability has on the degree of saturation of atmospheric tracers.

In 228-231: What is the proposal here? The notion that preformed O₂ is not always at equilibrium is hardly new. The problem is that better estimate of AOU would require knowing this disequilibrium component---which is highly variable and thus not trivial.

Yes indeed, several studies have pointed out the disequilibrium of oxygen during water mass formation and quantified it in more detail (e.g. Ito et al., 2004, Khim and Körtzinger, 2010, Olsen et al., 2010). We only want to illustrate how valuable measurements during active convection are in terms of estimating these undersaturated values for oxygen, which then again, for tracer studies further downstream or methods relying on exact estimated preformed oxygen values can not be neglected.

In 242: I would change "storage rate" to "rate of accumulation".

At this point we disagree. The term 'storage rate' is widely used in the literature to describe the rate at which anthropogenic carbon is added to the ocean, so we decided to leave it that way.

In 247: How much of the increased rate of accumulation can be attributed to rising atmospheric CO₂ versus interannual variation in circulation?

Thank you for that question. We calculated the expected anthropogenic carbon concentration based on rising atmospheric CO₂ concentrations only (Steinfeldt et al., 2009). By assuming an exponential increase of dissolved CO₂ in the surface mixed layer and using equilibrium carbon chemistry thermodynamic equations, the increase in anthropogenic carbon is expected to be 1.9% per year. We used these concentration to estimate expected inventories and added these estimates to Figure 4c. It shows that in the early 2000s, change in inventory is smaller than the expected value, meaning that the driver for the total inventory change is the circulation, as previously shown by Steinfeldt et al. (2009). In the late 2000s however, the increase in inventory is larger than the inventory based on atmospheric CO₂ increase only. That shows how important deep convective events are in order to sequester anthropogenic carbon.

REVIEWERS' COMMENTS:

Reviewer #1 (Remarks to the Author):

I addressed the points above in a previous review, and had two main clarifications which the authors have now addressed: a more complete discussion of the error on the saturation estimates, and some clarification on the buoyancy frequency calculation. I am satisfied with the author's revisions; In particular, I find the new discussion on buoyancy frequency and stratification greatly improved, with the author's interpretation now clear and supported by the figure.

I recommend the paper be accepted.

Point of clarification:

L39. Suggest adding what depths you are considering to be intermediate. 1000-2500 m?

Minor corrections (run-on sentences):

L35. Run-on. Break sentence: "such a location, however, the processes" -> "such a location. However, the processes"

L143-146. Run-on sentence. Suggest: "cannot explain our observations. Hence, other mechanisms"

L201-203. Run-on sentence. Suggest "substantiates this. The water column" or "substantiates this: the water column".

Suggestions only - these may be style rather than necessary, so please leave them out if you disagree.

L39-42. Long sentence. Suggest: "This decline in oxygen likely resulted from reduced exchange between the surface mixed layer and intermediate ocean, associated with warming and freshening in the upper ocean".

L53. Suggest "Despite the efforts" -> "Despite efforts"

L62. Suggest "this specific problem is now" -> "the problem of seasonal biases in sampling is now"

L71. Suggest "such a deep convective event" -> "in situ observations of deep convection"

L107-111. This is sort of an orphan paragraph. Include within another somehow?

L199. Suggest "there is as noted above"-> "there is, as noted above," or just "there is"

L283-286. Suggest "On a broader scale, the variations in ventilation (processes?) that are being uncovered may also help elucidate the decadal variability.." The intermediate clause "attributed with our growing observational data base" complicates the sentence, perhaps unnecessarily.

Reviewer #3 (Remarks to the Author):

I feel that the authors have adequately addressed my comments.

Final revisions:

Reviewer 1 (Remarks to the Author):

I addressed the points above in a previous review, and had two main clarifications which the authors have now addressed: a more complete discussion of the error on the saturation estimates, and some clarification on the buoyancy frequency calculation. I am satisfied with the author's revisions; In particular, I find the new discussion on buoyancy frequency and stratification greatly improved, with the author's interpretation now clear and supported by the figure.

I recommend the paper be accepted.

Point of clarification:

L39. Suggest adding what depths you are considering to be intermediate. 1000-2500 m?

Thank you for pointing this out. Helm et al. (2011) observe the describe changes in oxygen between 100-1000m. Hence, the term 'intermediate' ocean might be misleading, therefore we changed it to 'upper' ocean and ocean 'interior'.

Minor corrections (run-on sentences):

L35. Run-on. Break sentence: "such a location, however, the processes" -> "such a location. However, the processes"

Done.

L143-146. Run-on sentence. Suggest: "cannot explain our observations. Hence, other mechanisms"

Done.

L201-203. Run-on sentence. Suggest "substantiates this. The water column" or "substantiates this: the water column".

Done.

Suggestions only - these may be style rather than necessary, so please leave them out if you disagree.

L39-42. Long sentence. Suggest: "This decline in oxygen likely resulted from reduced exchange between the surface mixed layer and intermediate ocean, associated with warming and freshening in the upper ocean".

Done.

L53. Suggest "Despite the efforts" -> "Despite efforts"

Done.

L62. Suggest "this specific problem is now" -> "the problem of seasonal biases in sampling is now"

Done.

L71. Suggest "such a deep convective event" -> "in situ observations of deep convection"

L107-111. This is sort of an orphan paragraph. Include within another somehow?

L199. Suggest "there is as noted above"-> "there is, as noted above," or just "there is"

Done.

L283-286. Suggest "On a broader scale, the variations in ventilation (processes?) that are being uncovered may also help elucidate the decadal variability.." The intermediate clause "attributed with our growing observational data base" complicates the sentence, perhaps unnecessarily.

Reviewer 3 (Remarks to the Author):

I feel that the authors have adequately addressed my comments.